# Coherence in carotenoid-to-chlorophyll energy transfer

Elena Meneghin [1], Andrea Volpato [1], Lorenzo Cupellini[2], Luca Bolzonello[1], Sandro Jurinovich[2], Vincenzo Mascoli[2], Donatella Carbonera[1], Benedetta Mennucci [2] & Elisabetta Collini [1]

The subtle details of the mechanism of energy flow from carotenoids to chlorophylls in biological light-harvesting complexes are still not fully understood, especially in the ultrafast regime. Here we focus on the antenna complex peridinin–chlorophyll a–protein (PCP), known for its remarkable efficiency of excitation energy transfer from carotenoids—peridinins—to chlorophylls. PCP solutions are studied by means of 2D electronic spectroscopy in different experimental conditions. Together with a global kinetic analysis and multiscale quantum chemical calculations, these data allow us to comprehensively address the contribution of the potential pathways of energy flow in PCP. These data support dominant energy transfer from peridinin $S_2$ to chlorophyll $Q_y$ state via an ultrafast coherent mechanism. The coherent superposition of the two states is functional to drive population to the final acceptor state, adding an important piece of information in the quest for connections between coherent phenomena and biological functions.

[1] Department of Chemical Sciences, University of Padova, via F. Marzolo 1, 35131 Padova, Italy. [2] Dipartimento di Chimica e Chimica Industriale, University of Pisa, via G. Moruzzi 13, 56124 Pisa, Italy. Correspondence and requests for materials should be addressed to E.C. (email: elisabetta.collini@unipd.it)

   1

Dynamics of energy transport in light-harvesting complexes is traditionally attributed to the peculiar properties of tetrapyrrole compounds, mainly chlorophylls (Chls) and bacteriochlorophylls. A particular family of marine organisms of the dinoflagellates phylum has instead evolved light-harvesting machinery, mainly relying on carotenoids (Cars) as major harvesting pigments. Their external antenna, the peridinin–chlorophyll a–protein (PCP), also collects light in the blue-green spectral region of the visible spectrum, where solar radiation peaks below sea level, and where Chls do not absorb (Fig. 1). In fact, the early publication of PCP high-resolution X-ray structure[1] revealed that this water-soluble protein has a pseudo-two-fold symmetry and binds eight peridinins (Pers) clustered around two Chl molecules. The detailed structural information, together with the remarkable efficiency for excitation energy transfer (EET) from Car donors to Chl acceptors (~90%)[2–5], have inspired intense research among the biophysical community[6–12]. While the properties of Cars as leading actors in photoprotection[13] and non-photochemical quenching[14] have been the object of several in-depth investigations in the last decades[15–17], the mechanistic and dynamic details of the excitation energy flow from Cars to Chls are still not fully understood. One of the main difficulties is a limited knowledge of the electronic properties of Cars[18–21]. PCP, with its high content of Cars, represents therefore an ideal test case. Moreover, the presence of Pers, peculiar Cars substituted with polar groups, may lead to intriguing features.

Traditionally, Cars' electronic properties are described with a three-level model, including a ground state, $S_0$ $(1^1A_g^-)$, and two low-lying singlet excited states, $S_1$ $(2^1A_g^-)$ and $S_2$ $(1^1B_u^+)$. It follows that $S_0 \rightarrow S_1$ transition is symmetry-forbidden and that $S_2$ is the first bright state[22]. Early femtosecond spectroscopy experiments on PCP[3–5] revealed that after photoexcitation, $S_2$ state rapidly relaxes to $S_1$ state via internal conversion in tens–hundreds of femtoseconds. These experiments led to models in which the excitation energy was mainly funneled from the $S_1$ state of Per to the $Q_y$ state of Chl with a characteristic time constant of 2.5–3.5 ps[8].

However, the weak oscillator strength for the $S_0 \rightarrow S_1$ transition and the remarkable efficiency of EET process toward Chl appeared immediately contradictory. Therefore, also considering the presence of polar groups in both the backbone and the terminal rings of Per, the formation of an intramolecular charge transfer (ICT) state has been proposed. The ICT state was

described as intimately coupled to the dark $S_1$ state, and crucial for its capability to enhance the $S_0 \rightarrow S_1$ transition dipole moment, leading to a more efficient EET between Per and Chl[4].

Like for other antenna complexes bearing Cars, femtosecond spectroscopic experiments on PCP have also suggested the presence of ultrafast channels of EET from $S_2$ to the Q bands of Chl[4,7–9,23] ($Q_x$ and/or vibronic states of $Q_y$), also with possible contributions of coherent pathways[24,25]. However, clear and direct evidence is still lacking to identify the main pathways of energy flow from Pers to Chls, and their timescales, including the role of vibrational modes on the dynamics.

Starting from the existing body of preliminary knowledge already available and having in mind the questions still open about the mechanism of EET, the role of $S_2$ state in EET process in PCP is investigated. We also exploit the higher spectral resolution and the higher dimensionality of two-dimensional electronic spectroscopy (2DES) to relieve the spectral congestion of the multi-chromophoric system of the protein complex. Guided by multiscale quantum chemical simulations, 2DES spectra are acquired in different experimental conditions, exploiting the spectral filtering action of the exciting bandwidth to turn on, or off, from time to time the contributions of Pers. This procedure, together with high-level data analysis methodologies, allows a clear identification of the multiple channels of energy flow from Per to Chl with an unprecedented level of detail. The associated timescales are characterized, and the identification of important mechanistic details, including the presence of coherent transfer and the potential involvement of vibrational modes of Per, is assessed.

## Results

**Prediction of the excitonic spectrum and rate analysis.** Ultrafast spectroscopic characterization was guided by prior simulation of the excitonic spectrum of Cars and by the calculation of EET rates as a function of Pers energies. It is now known that the particular three-dimensional arrangement of Pers in PCP promotes the formation of excitonic states partially delocalized on subsets of Pers. Accounting for such a delocalization and for the correct couplings between Pers and Chls is crucial to predict the correct energies of the states involved in the energy flow and associated rates. Quantum chemical calculations have been performed to simulate the excitonic spectrum of the complex, including the $S_2$ bright states of the eight Pers and the Q and Soret (B) bands of the two Chls, resulting in 16 total states. The electronic couplings among all excited states were computed with an approach based on the transition densities of the interacting pigments. All calculations also include the effect of the protein described as a classical polarizable embedding (see Methods and Supplementary Methods 1). The good agreement between simulated and experimental spectra is proved by the comparison shown in Fig. 1.

The calculations predict that EET from $S_2$ toward $Q_y$ state is much faster than toward $Q_x$ state, due to the larger couplings between $S_2 \rightarrow Q_y$ states (Supplementary Fig. 4, Supplementary Table 5). According to that, the EET $S_2 \rightarrow Q_x$ channel is unlikely to show a noteworthy efficiency, disproving what was suggested in previous works[4,7–9,23]. From the quantum chemical calculations, we predict that the fastest EET channel from Per to Chl likely involves a specific pair formed by Per624 and Chl602 (Fig. 2d). This pair is characterized by a large electronic coupling due to the favorable relative orientation and distance between the two pigments (Supplementary Table 5). Moreover, Per624 contributes to the red edge of the Pers absorption spectrum due to its largely red-shifted site energy. The higher red-shift of Per624 can be explained in terms of electrostatic and polarization

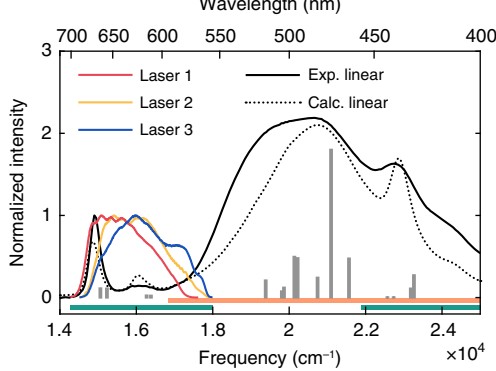

**Fig. 1** PCP linear absorption spectrum. Experimental spectrum (black solid line) in sodium phosphate buffer (pH 7.5) and laser spectrum profiles (colored lines) used in the three 2DES experiments. Calculated excitonic spectrum (black dotted line) and corresponding positions of the excitonic states (gray bars). Orange and green bars show the absorption regions of Pers and Chls, respectively

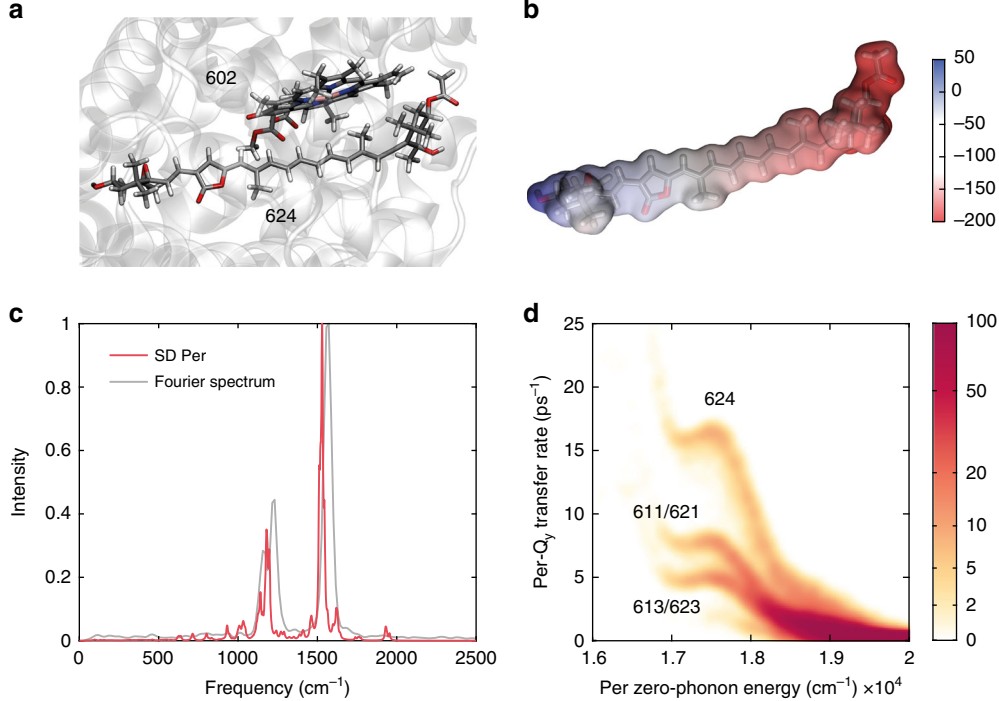

**Fig. 2** Results of quantum mechanical simulations. **a** Structure of the Per624–Chl602 pair. **b** Electrostatic potential on the molecular surface of Per624 due to the protein environment. **c** Average spectral density calculated for the Pers (red) compared with the Fourier transform of the oscillating part of the 2DES signal recorded with laser 3 (gray) (see also Supplementary Fig. 1). **d** Calculated two-dimensional heat map of the correlation between the zero-phonon energy of the Per exciton (x-axis) and the $S_2 \to Q_y$ EET rates (y-axis). The color bar represents the number of occurrences. Some regions of the heat map are labeled with the Per, which the exciton is localized on. The numeric labels pinpoint distinct chromophores as described in ref.[1]

effects of specific residues placed around the terminal rings (Supplementary Fig. 2 and 3). These induce a strong electric field (Fig. 2b) oriented along the main axis of the pigment, thus largely stabilizing the $S_2$ state. Additional details on this analysis are reported in Supplementary Note 1.

Of even further interest, the $S_2 \to Q_y$ calculated transfer rates for the pair Per624–Chl602 show a nontrivial probability distribution, close to a lognormal distribution (Supplementary Fig. 5). This has a tremendous effect, particularly for the interpretation of the experimental data. Indeed, while in the familiar normal distribution models, energy transfer is described by a single exponential function with a time constant equal to the center of the distribution, here the time parameter must be represented by a probability density function, as suggested by other authors in similar systems[26,27].

**Population dynamics.** With the aim of capturing the details of the fastest EET channel, the experimental conditions for the 2DES measurements have been set according to these theoretical predictions and, thus, the laser pulses have been tuned to the red edge of the linear absorption spectrum of Per, as shown in Fig. 1. Three different experiments were performed. In each, the exciting bandwidth was carefully modulated with pulse-shaping techniques to act as a spectral filter[28–30] and turn off the contribution of the Per donor (laser 1), or progressively brought into resonance with Per (laser 2 and 3), to follow the effects of the coupling between Per and Chl.

The 2DES experiment with laser 1, detuned from the $S_2$ absorption band, was taken as a control measurement to study the 2DES response of Chl in PCP. Absorptive 2DES maps recorded in this condition (Fig. 3a) reveal a behavior in full agreement with previous 2DES characterization of Chl solutions[31–34]. Indeed, the maps are dominated by an intense positive

diagonal peak centered at $14\,850\,cm^{-1}$ (position A in Fig. 3b), easily assigned to ground-state bleaching (GSB) and stimulated emission (SE) of the $Q_y$ state. The weaker cross-peak at $(16\,000, 14\,850)\,cm^{-1}$ (position B) can be attributed to the coupling of the $Q_y$ electronic transition with vibrational modes between 1000 and $1200\,cm^{-1}$, and to an ultrafast internal conversion process from $Q_x$ to $Q_y$ states, as detailed in previous works[33,34].

2DES maps recorded at 100 fs with laser 2 (Fig. 3b) and 3 (Fig. 3c) reveal the presence of a strong negative feature at excitation energy $> 17\,000\,cm^{-1}$ (position C). This is due to excited-state absorption (ESA) signal from the Per $S_1$ state that obscures, almost completely, the GSB- and SE-positive signals from $S_2$ state. The rise of this strong signal in the ultrafast nonlinear response of PCP is widely documented in the literature[3–5,8] and reflects the presence of the $S_2 \to S_1$ internal conversion. In the experiments conducted with laser 3, together with the ESA from $S_1$, a rapidly decaying ESA from $S_2$ was also captured. Indeed, at early times (Fig. 3g), an additional negative feature is present at position D $(17\,500, 17\,000)\,cm^{-1}$. This feature disappears in $<30$ fs, in agreement with the short lifetime of $S_2$. As discussed below, the dynamics of the main $S_1$ ESA signal is strongly influenced not only by the rate of the $S_2 \to S_1$ process but also by the presence of other ultrafast EET pathways affecting the population decay of $S_2$. This implies that although the strong negative ESA features may hide other important contributions, they still retain important information on the competing transfer mechanisms that deplete the $S_2$ state.

In fact, a detailed analysis of the signals at excitation frequencies $> 17\,000\,cm^{-1}$ under the three different exciting conditions unveiled important details on the Per $\to$ Chl EET. First, the trend shown by the three sets of 2DES measures disproves previous hypothesis that invoked the $S_2 \to Q_x$ pathway as responsible for the rise in the GSB signal in transient

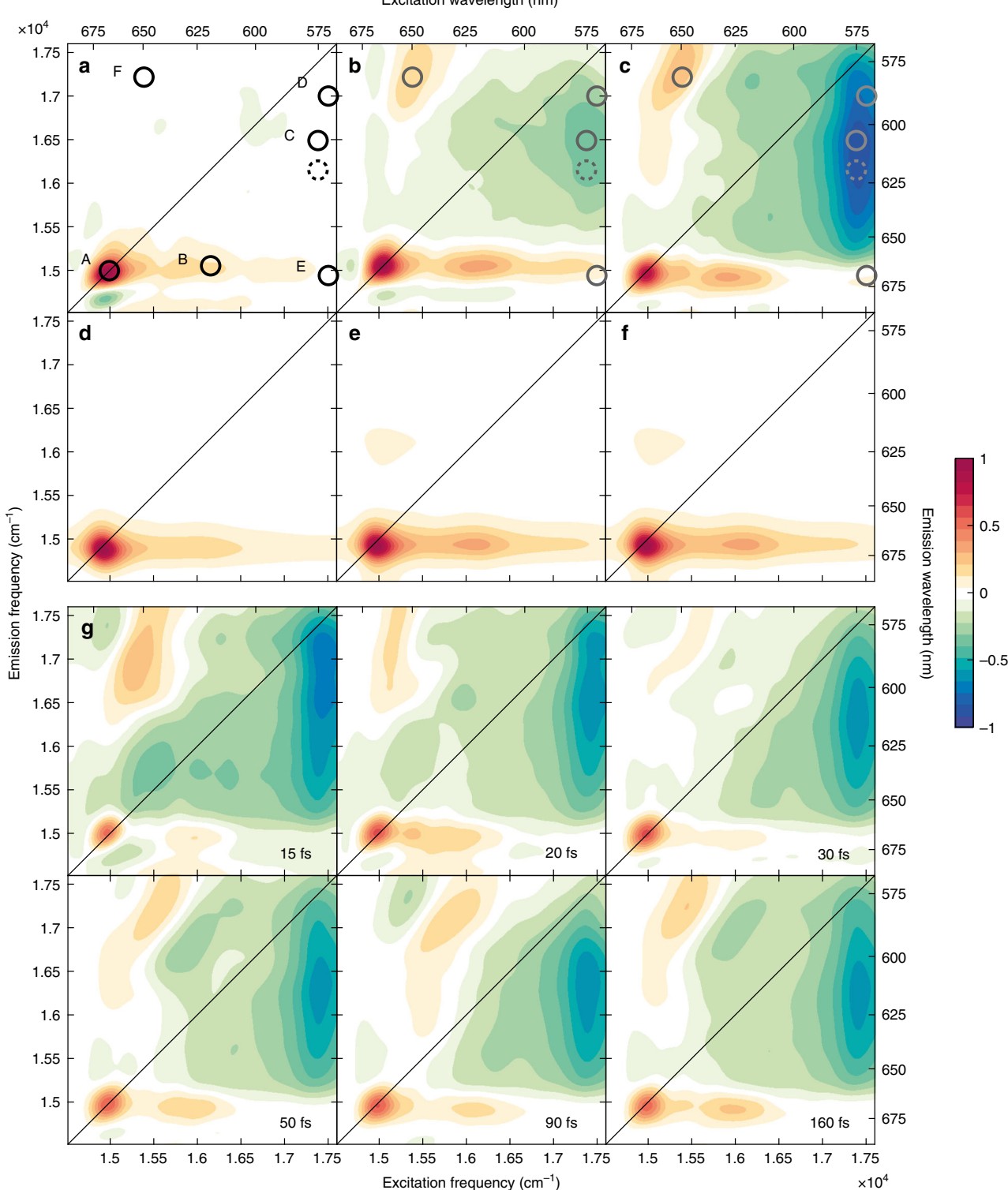

**Fig. 3** Absorptive 2DES signal of PCP at 295 K. **a–c** Experimental and **d–f** simulated 2DES maps at a selected value of population time (100 fs) obtained with laser 1 (**a**, **d**), laser 2 (**b**, **e**), and laser 3 (**c**, **f**), respectively. More details of experimental and simulated data are reported in Supplementary Fig. 11-13. Maps are normalized to 1 at their maximum. The dots labeled from A to F in **a** pinpoint relevant positions as commented in the main text for all the laser bandwidths. **g** 2DES maps recorded with laser 3 at different population times

absorption experiments[4,8,9,23]. Indeed, if $S_2 \to Q_x$ was the dominant pathway, the 2DES map would display a rising cross-peak at coordinates (17 500, 16 150) cm$^{-1}$, tentatively indicated with a dotted circle in Fig. 3a–c, but no rise at these coordinates was detected.

Even assuming that this signal is canceled by the negative ESA contribution, the dynamics of ESA at these coordinates would have maintained some features of this pathway, detectable through kinetic models. This is not the case. First of all, no meaningful differences could be detected looking at the evolution

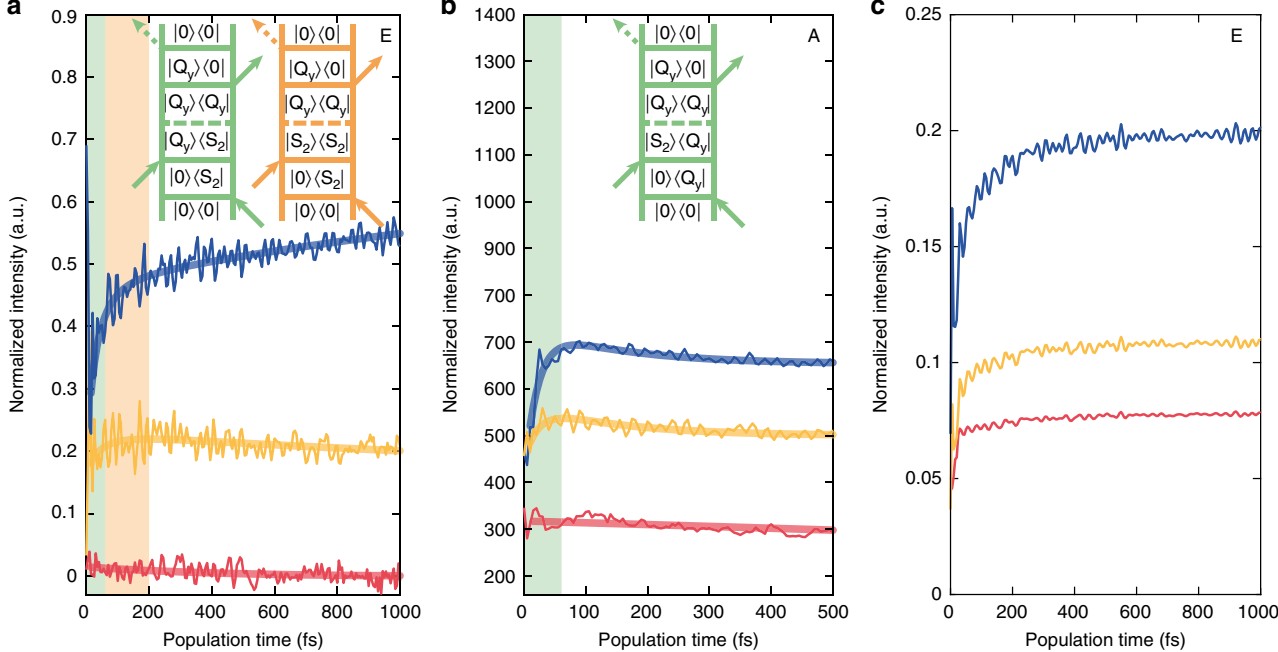

**Fig. 4** Experimental and simulated time traces extracted at relevant coordinates in the real absorptive 2DES maps. See points A and E in Fig. 3b. Red, yellow, and blue lines point out measures performed with laser 1, 2, and 3, respectively. **a** Experimental data (thin lines) and results of the global kinetic analysis (thick lines) at cross-peak coordinates E, (17 500, 14 850) cm$^{-1}$. Laser 2 and 3 traces are shifted by 0.2 and 0.4, respectively, to help visualization. Feynman diagrams describing the EET pathways giving rise to signal in E are also shown. **b** Experimental signal (thin lines) and fitting traces (thick lines) integrated over the diagonal $Q_y$ signal at coordinates A (14 850, 14 850) cm$^{-1}$. Feynman diagram describing the rising signal in A. Traces recorded with laser 2 and 3 are shifted by 200 and 400 units, respectively. **c** Simulated real absorptive signal at cross-peak E

of the 2DES trace along $t_2$ extracted, where the $S_2/Q_x$ peak should show up and where only $S_1 \rightarrow S_n$ ESA contribution is expected.

Moreover, we verified that the decay of $S_2$ population matches the rise of $Q_y$ (Supplementary Fig. 14). Indeed, we identified a clear rising signature at (17 500, 14 850) cm$^{-1}$ (position E in Fig. 3). The control measurement (laser 1) did not show this rise that, instead, became greater as we switched from laser 2 to laser 3 (Fig. 4a). The particular frequency coordinates, suggest the attribution of this feature to an increasing signal due to $S_2 \rightarrow Q_y$ EET.

At least two possible mechanisms of EET can be identified, as represented by the Feynman diagrams shown in Fig. 4. Together with the common incoherent population transfer pathway, a coherent mechanism can also be envisioned, where $S_2/Q_y$ coherence prepared by the laser pulse after the second interaction dephases during the population time and gets localized on the $Q_y$ population (panel a). Assuming that this second coherent mechanism is active, we would also expect a concomitant rise of the signal at diagonal position A, witnessing an increase of the $Q_y$ population following the transfer from $S_2$, as described by the diagram in panel b. Indeed, this behavior is clearly visible with laser 3, it is subtle with laser 2 and it is not present in the control measurement with laser 1 (Fig. 4b). It should be noted that this rise cannot be justified with the conventional diagrams describing GSB and SE of the $Q_y$ band. Additional evidence supporting the presence of coherent transfer will be provided by the beating analysis.

The next step is to quantitatively determine the rates of the transfer through a fitting of the rising traces recorded with laser 3. Owing to the multiplicity of possible pathways and the spectral congestion in the examined spectral region, a global fitting procedure based on a specific kinetic model was adapted from a previously proposed global analysis method[35]. This method

allows for performing a global fitting of the whole set of 2DES data using, as fitting functions, the solutions of a suitably defined kinetic scheme. This approach is much more convenient than the usual multi-exponential fitting, as it allows treating arbitrary kinetic schemes (Supplementary Methods 2).

The best fitting results were obtained from the kinetic model shown in Fig. 5a, where the time constants regulating the relaxation process are also highlighted. Following calculations, the rate of $S_2 \rightarrow Q_y$ process was described in terms of a lognormal distribution: $\rho(t|\mu,\sigma) = 1/(t\sigma\sqrt{2\pi})\exp\{-(\ln t - \ln\mu)^2/(2\sigma^2)\}$; with $\ln\mu$ mean value and $\sigma$ standard deviation of the distribution[36]. This implies that, for the $S_2 \rightarrow Q_y$ process, a probability density function rather than a single time constant emerges from the fit. Applying this fitting model, we found $\mu = 63$ fs and $\sigma = 0.985$ (Fig. 5c). This process is competitive with the $S_2 \rightarrow S_1$ internal conversion (>150 fs estimated time constant). On a longer timescale, energy is transferred to $Q_y$ through the $S_1 \rightarrow Q_y$ channel (~2 ps).

Figure 5b summarizes the time evolution of the states involved in EET, according to this fitting model. It should be noted that in the time and spectral window investigated, the population of the final acceptor $Q_y$ state, although fed by two distinct channels, derives mainly from $S_2$.

Recently, another kinetic scheme has been proposed to describe EET from $S_2$ to $Q_y$, which relies on a preliminary step of relaxation of the $S_2$ state geometry from the Franck-Condon to a distorted conformation[24]. Our quantum chemical calculations, however, seem to rule out the presence of such a preliminary step as there is no trace of relaxed twisted configurations of Per in the $S_2$ state.

As shown in Fig. 4b, specific information on the coherent regime of EET can be extracted from the dynamics of the diagonal peak, whose initial ultrafast rise has already been

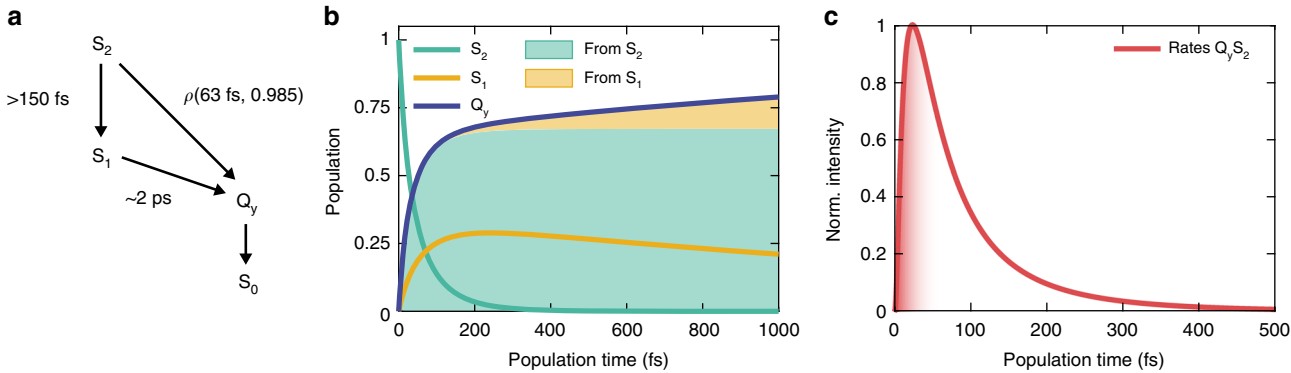

**Fig. 5** Kinetic analysis. **a** Kinetic scheme of excitation energy transfer pathways in PCP. **b** Time evolution of populations according to the results of the fitting based on the kinetic model in **a**. **c** Lognormal distribution of $S_2 \rightarrow Q_y$ transfer rates. Shaded area suggests the time domain of coherent EET

attributed to a purely coherent EET pathway. This rise could be fitted with an average time constant of 22 fs, which can be used to estimate the timescale in which coherent EET is relevant, as shown in Fig. 5c.

**Oscillating contributions**. The evolution of the 2DES maps along the population time, together with the population decays already described, revealed the presence of a lively beating behavior, mainly due to the activation of the strongly coupled vibrational modes of the Per. In particular, strong components beating at 1165, 1230, and 1565 cm$^{-1}$ were detected. They matched the resonant Raman signals of Per in acetonitrile[37] and the calculated spectral density of the Per (Fig. 2c and Supplementary Fig. 16).

The predominance of Per vibrational modes in the 2DES response, with respect to Chl modes, only just identifiable from noise, is expected in the light of the large Huang-Rhys factors characterizing these modes.

To check for the presence of electronic coherences between Per and Chl states, whose relevance has been recently suggested in the literature[24,25], the beating behavior of the rephasing signal (recorded with laser 3) in the $S_2/Q_y$ cross-peak region has been carefully scrutinized, applying a recently proposed method based on a time-frequency transform (TFT) analysis[38,39]. Given an oscillating trace, this analysis allows extracting not only the main frequency components contributing to the beating, as in conventional Fourier transform, but also their time behavior. The result is a bi-dimensional time-frequency plot, such as the one reported in Fig. 6d. Here the frequency and dephasing times of the relevant components are identified on the $y$- and $x$-axes, respectively. The TFT analysis of the 2DES trace at these positions (Fig. 6a) unveils four main oscillating contributions. The first three components have frequencies of approximately 1150, 1210, and 1550 cm$^{-1}$ and dephasing times of several hundreds of femtoseconds. They correspond to the same vibrational modes of Per already identified in the analysis of the overall 2DES maps. The presence of Per vibrational coherences at these positions is in agreement with simulated data (Supplementary Fig. 16–19).

The fourth component is a quickly dampened oscillation centered at ~1900 cm$^{-1}$ (Fig. 6b–d), which dephases in ~20 fs. Its contribution was exhausted after 75 fs (panel c) and characterized by a large bandwidth, in agreement with the time-frequency indetermination principle. In the first tens of femtoseconds, this feature added to the longer-lived vibrational signals and produced negative amplitudes in Fig. 6d (Supplementary Fig. 9). More details on this investigation are reported in Supplementary Note 3. The same analysis performed at corresponding positions

above diagonal produced similar results. Although the dominating contributions of the Per vibrational modes hinder the direct detection of the fourth broad component, its presence is indirectly manifested through negative amplitude features (Supplementary Fig. 10). The frequency and time behavior of this beating component, together with the peculiar coordinates at which it contributes, point toward the attribution of these dynamics to an electronic coherence between Per $S_2$ and Chl $Q_y$.

In support of this interpretation, Fourier analysis (Fig. 6c) and time-frequency plots (Supplementary Fig. 7) generated from the traces at fixed emission frequency (colored dots in Fig. 6a) revealed that this ultrafast feature shifts accordingly to the excitation energy. Indeed, the sampling at increasing excitation energies results in the selection of Pers with slightly higher $S_2$ site energies. As a consequence, the coherent superposition with Chl $Q_y$ state would also beat at slightly higher frequencies.

The presence of the same beating frequency at the symmetrical position above the diagonal, expected for an electronic coherence[40], could not be ascertained as this portion of 2DES maps is strongly dominated by vibrational modes of Per. The strong oscillations above diagonal are clearly visible also in Fig. 3b, c, g as a large cross-peak around (15 500, 17 000) cm$^{-1}$ (position F in Fig. 3), strongly beating along the population time. The intensity of this contribution increased progressively with respect to the amplitude of the oscillations below diagonal, moving from laser 1 to laser 2 and 3, i.e., increasing the contribution of resonant $S_2$ states. These coherent signals have the frequency and the time behavior of Per vibrational coherences. However, they occur in a spectral region not predicted by the displaced harmonic oscillator model[40] for these modes, suggesting the presence of more complex mechanisms. It has been recently suggested that vibrational modes of accessory pigments could work as a sink, absorbing the excess energy released by energy transfer due to the mismatch of donor and acceptor levels[41,42]. In the case of PCP, the strongly beating signal in the upper-left portion of the maps would represent the deposition into the Per vibrational modes of the excess energy released after the $S_2 \rightarrow Q_y$ transfer. Further investigations are needed before definitive conclusions can be reached.

**2DES simulations**. All the experimental results described above can be framed in a unified picture with the help of theoretical simulations. Simulations of the 2DES signal are performed following the standard approach based on the third-order perturbation theory[43,44], where a response function is fully convoluted with all the exciting pulses[45]. The simulation of the third-order optical response was fed with parameters obtained with the

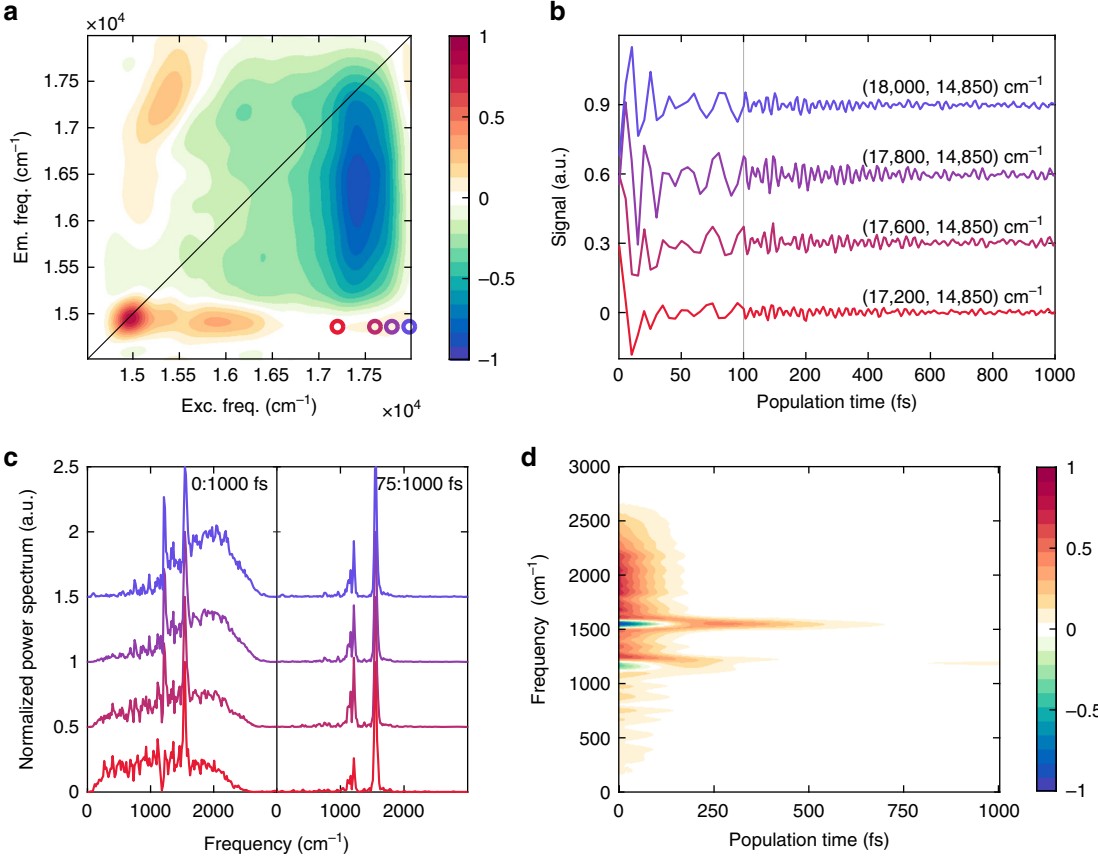

**Fig. 6** Analysis of the beating behavior of the $S_2/Q_y$ cross-peak. **a** The coordinates at which the analysis has been performed are pinpointed as colored dots in a 2DES map. **b** Time traces at the corresponding coordinates. Note the break in the time axis at 100 fs. **c** Fourier transform (FT) of the traces in **b** computed in the time windows (0:1000) fs (left) and (75:1000) fs (right), confirming that the broad component centered at about 1900 cm$^{-1}$ is a quickly dephasing signal and not just a FT artifact. **d** Time-frequency transform of the trace at (17 800, 14 850) cm$^{-1}$

quantum chemical calculations described above; here, however, only the excited states $Q_y$ of Chl602 and $S_2$ of Per624 were considered for the sake of simplicity. Calculations indeed suggested that this pair provides the major contribution to the faster EET channel captured by the experimental data and therefore this choice allows simplification of the calculations without losing in generality.

Inhomogeneous broadening has been included, averaging over 200 realizations, and the real spectral profile of the exciting pulses has also been considered. Examples of simulated 2DES maps at the three different excitation conditions are reported in Fig. 3d–f. Clearly, the simulated maps could not reproduce the strong ESA signal recorded in the experiments, as the $S_1$ state is not included in the simplified model. Nevertheless, the simulations are able to capture the $Q_y$ main features and the intertwined dynamics between $Q_y$ and $S_2$, supporting the main findings of the work. The relative ratio between the main diagonal peak of Chl at coordinates A and the cross-peak at coordinates B complies with the experimental data and it is a consequence of the filtering effect of the different laser spectra. Despite the overwhelming contribution of ESA in the experiment, a notable correspondence between experimental and simulated traces of the $S_2/Q_y$ cross-peak dynamics at A position (Fig. 2) has been found, as shown in Fig. 3c.

Also, the coherent dynamics and the vibrational oscillating patterns were fully reproduced, with the only exception of the strong beating feature on the upper diagonal portion of the maps. Again, this confirms the non-conventional origin of this feature.

## Discussion

We have reported here 2DES measures recorded in different experimental conditions, analyzed with state-of-the-art data analysis methods, and supported by multiscale quantum chemical calculations on the pigment–protein complex. These measures allowed a thorough and complete mapping of the multiple pathways mechanism of energy flow in PCP, and the relative time constants. The picture emerging from computational and experimental data is that contrary to that generally assumed, the energy flow from Per to Chl in the investigated time and spectral windows, is dominated by a direct channel involving $Q_y$ state of Chl as acceptor and the $S_2$ state of the Per as donor. These findings represent a further insight toward the full resolution of a long-term discussion about the effective states involved in the energy flow from Per to Chl. More importantly, the fast components within the distribution of EET rates, which are characterized by an average time constant of approximately 63 fs, seem to be notably affected by coherent mechanisms. These include a nontrivial sink effect by vibrational modes of the Per, which may contribute to the high efficiency of this relaxation pathway. In fact, we recorded the evolution of the coherent superposition of $S_2$ and $Q_y$ detected as a beating frequency with ~20 fs dephasing time. The evolution of this coherence, prepared by the laser pulse, appears functional to the EET, since a steep rising of the $Q_y$ diagonal signal, justifiable only with coherent pathways, has been recorded within the same timescale. This suggests that the evolution of the coherence effectively moves the population on the $Q_y$ state. Our data and calculations suggest that

in the most coupled Per–Chl pair, partially delocalized states are created upon photoexcitation and they dephase quickly, localizing population on the final acceptor state. Thus, the presence of coherence appears functional to the energy distribution in PCP. This is important information in current efforts to evaluate possible opportunities to harness coherence to realize, control, and/or drive energy transduction[46].

While our results here refer to a specific protein, nevertheless they demonstrate that the transfer from Cars to Chl can sustain coherent dynamics also when the spectral detuning seems to be not fully favorable. This is an important piece of information in the current efforts of unveiling the still unclear mechanism of energy flow from Cars to Chl.

## Methods

**Two-dimensional electronic spectroscopy**. We performed 2DES experiments in the fully non-collinear BOXCARS geometry, employing the setup described in ref.[47]. The 2DES experiments were repeated in three different experimental conditions: the laser spectrum was centered at 15 600 (laser 1), 15 900 (laser 2), and 16 200 cm$^{-1}$ (laser 3), respectively, with ~2000 cm$^{-1}$ full-width at half maximum. The pulse duration was determined by FROG measurements at the sample position: the broadband pulses were compressed to 11.0, 8.9, and 8.5 fs with laser 1, 2, and 3, respectively (Supplementary Fig. 15). They were attenuated to 5 nJ per pulse by a broadband half-waveplate/polarizer system. All measurements were performed under room temperature (295 K). Population time was scanned up to 1 ps with 5 fs time step and coherence time from −80 to 100 fs in 1 fs steps. Each experiment was repeated and averaged three times to ensure reproducibility.

Data analysis was performed exploiting the global fitting methodology described in ref.35 using a kinetic model to detail the populations dynamics. The first 10 fs have been omitted in the analysis to avoid pulse overlap artifacts.

**Sample preparation**. PCP antenna complex was extracted and purified by Professor R. G. Hiller from *Amphidinium carterae*, according to the method previously published in ref.48. For the 2DES experiments, the sample was diluted in sodium phosphate buffer (pH 7.5) until an optical density of about 0.4 in a 1 mm cuvette was reached in the region of the $Q_y$ band.

The sample was accurately degassed under nitrogen flux to remove the oxygen from the solution and the cell was sealed immediately after to avoid the formation of oxidized species during 2DES measurements.

**Quantum chemical calculations**. The crystal structure of PCP[1] was refined by relaxing the geometries of the pigments through a quantum mechanics/molecular mechanics (QM/MM) optimization in a frozen protein environment. The excited states of the complex are described through an exciton model[49] that comprises the $S_2$ bright states of the eight Pers, plus the Q and B states of Chls, resulting in 16 total states. The electronic couplings among all excited states were computed with an approach based on the transition densities of interacting pigments[50]. All excited-state calculations were performed using a multiscale polarizable QM/MM scheme (QM/MMPol)[50]. Within this framework, the protein (and the other cofactors) surrounding the pigment of interest are described through a set of atomic point charges and isotropic polarizabilities. The QM/MMPol scheme accounts for mutual polarization effects between the pigments and the environment; moreover, it allows the environment to respond to the excitation of the pigment and to properly describe the dielectric screening due to the anisotropic environment of the protein[50] in the calculation of the electronic couplings. For all calculations, we used the (TD)-DFT method with the B3LYP[51] functional.

Spectra and dynamics were modeled with the disordered exciton model, using the second-order cumulant expansion with modified Redfield theory[52]. Further details are presented in the Supplementary Methods 1.

**Theoretical 2DES spectra**. The 2DES signal of PCP was simulated using the excitonic heterodimer model reported in ref.53. Excited-state $Q_y$ of Chl602 and $S_2$ of Per624 were considered, exerting an electronic coupling of 308.4 cm$^{-1}$ computed in the full excitonic model. The simulation of the third-order optical response was fully fed with parameters computed in the quantum chemical calculations. The strong inhomogeneous broadening of Per system asked for an averaging of the optical properties over 200 disorder realizations. Each snapshot was characterized by specific site energies, transition dipole moments, lineshape functions, and transfer rates (Supplementary Notes 1 and 2). Rephasing and non-rephasing impulsive contributions were fully convolved with the experimental laser spectra[45], in order to match the experimental signals.

**Data availability**. All data supporting the findings of this study are available from the corresponding author upon request. All the custom codes used in this study are available from the corresponding author under reasonable request.

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

## Acknowledgements

This work is financially supported by the ERC Starting Grants QUENTRHEL (278560) and ENLIGHT (277755). We thank Professor R.G. Hiller (Macquarie University) who kindly provided the PCP samples used in this work.

## Author contributions

E.M., A.V. and L.B. performed spectroscopic measurements and data analysis; L.C., S.J. and V.M. performed QM calculations; E.C. and B.M. designed the research; E.C., B.M., and D.C. helped with data interpretation. E.M., A.V., E.C., L.C. and B.M. wrote and edited the manuscript. All authors reviewed and discussed the manuscript.

## Additional information

**Competing interests:** The authors declare no competing interests.

