## [Peer Review File · Nature Communications]

Reviewer #1 (Remarks to the Author):

General Impression

This is a well-written manuscript with a very careful and thorough analysis, which demonstrates the huge amount of physical insight gained from interplay of computational and careful experimental work. The figures were extremely clear and well labelled, supporting the claims made in the main text.

Main Idea(s) / Summary

A combination of careful experimental execution and global fitting analysis of 2DES spectra obtained using three different laser bandwidths with quantum chemical and quantum dynamical simulations elucidated electronic energy transfer (EET) within peridinin – chlorophyll-a – protein (PCP). It was found that EET from peridinin (Per) to chlorophyll-a (Chl-a) mainly involves the collapse of the electronic coherence between the Per S₂ and Chl-a Q_y states, in which more than two-thirds of the initially prepared Per S₂ is transferred. This electronic coherence dephases on sub-30 fs timescale and is deduced from a global analysis of experimental data. This conclusion debunks prior observations that the dominant energy transfer pathway is the Per S₁ → Chl-a Q_y state.

Discussion / Critical Analysis / Comments for the Authors

1. I like how you have used time-frequency analysis to tease out the 1900 cm⁻¹ electronic coherence from signals around the below-diagonal Per S₂/Chl-a Q_y cross peak obtained with laser 3. May you elaborate further on the difficulty of applying the same analysis to the corresponding above-diagonal cross peak, since the time-frequency analysis should be able to tease out the Per vibrational mode contribution?
2. Shouldn't the electronic coherence be closer to 2250 cm⁻¹, since the lowest Per S₀ – S₂ transition occurs around 17250 cm⁻¹ and the Chl-a Q_y transition at 15000 cm⁻¹ from the 2DES spectra shown in Figure 3 and Figure S13?
3. Is there a reference or citation for the quantum chemical simulations that supports eliminating the presence of the relaxed twisted configuration of Per in the S₂ state?

Reviewer #2 (Remarks to the Author):

This manuscript presents a combined experimental and theoretical study on energy transfer dynamics in peridinin-chlorophyll *a*-protein (PCP), focusing in particular on the energy flow from the bright S₂ state of peridinin (Per), the principal light harvester in PCP, to the low-lying Q states of chlorophylls (Chls). By using ultrafast two-dimensional electronic spectroscopy, the authors identify a Per S₂-Chl Q_y cross peak with a 63 fs energy transfer rate, proposing the S₂→Q_y pathway as the dominant energy transfer pathway in PCP instead of S₂→Q_x, as has been reported in the literature. In addition, signatures of potential electronic coherence between Per S₂ and Chl Q_y are observed, which appears consistent with the observed very rapid (63-fs) energy transfer rate.

PCP, found in the light-harvesting machinery of marine dinoflagellates, exhibits distinct photophysical properties from the light-harvesting complexes of bacteria and plants: 1) it uses carotenoids (peridinins), not chlorophylls, as the principal light harvester, and 2) peridinin is a highly substituted carotenoid, giving rise to unusual behaviors, e.g. strong dependence on solvent polarity, presence of an intramolecular charge transfer state. These unique characteristics of PCP make it an intriguing system to elucidate the dynamics of Car-Chl energy transfer, specifically the photophysical role of peridinins, which is still not completely known. Furthermore, the experiments employ advanced spectroscopic technique and are paired with cutting-edge theoretical work. However, the two major conclusions (direct S₂→Q_y energy transfer and role of coherence in energy transfer) both require further substantiation, as described in more detail below. Without additional conclusions or justification, this manuscript is more appropriate for a field-specific journal.

Major comments:

1. Page 5, last paragraph: the authors state that they observe no S₂-Q_x cross peaks, which leads them to propose Per S₂→Chl Q_y as the exclusive energy transfer mechanism between Per and Chl. This is reasonable given the significantly smaller coupling between S₂-Q_x compared to S₂-Q_y, but also surprising considering the poor spectral overlap as well as larger energy gap between Per S₂ and Chl Q_y. Furthermore, if energy transfers from S₂ to Q_x on similar or slower timescale than from Q_x to Q_y, no peak would appear. Thus, a more detailed analysis might be worthwhile, such as:

a. To rule out inverted kinetics as described, the authors should show the decay of S₂ matches the rise of Q_y.

b. To rule out the presence of an S₂ peak, while it clearly looks like the major feature is the intense negative S₁→S_N excited-state absorption (ESA) of Per₆₂₄ that is masking all the positive features, if there is a visible S₂-Q_x energy transfer channel, monitoring the difference in the waiting time traces between where the S₂-Q_x peak should show up and where there is only Per S₁ ESA may unmask the kinetics of S₂-Q_x energy transfer.

2. The authors describe the ~22 fs processes observed as arising from coherence between Per S₂ and Chl Q_y. In the case of ultrafast excitation, however, the defined phase relationship between pulses one and two excites a coherence, which decoheres rapidly (<50 fs timescale) due to the small couplings between the pigments. These oscillations emerge from ultrafast excitation, and carry no information about energy transfer dynamics. These behaviors are described theoretically in the literature, for example Schlau-Cohen, et al. *Nature Chem* 2012 4, 389.

3. A final major issue I have with this work is that only one specific energy transfer pathway (Per624→Chl602) is directly interrogated due to limited bandwidth of the NOPA spectrum. This is only a single pair of Per and Chl out of the many donor-acceptor pairs that can be formed with 4 Per's and one Chl per subunit of PCP. The NOPA spectra employed in this work may be beneficial in that they allow for selective (de)excitation of Per624, as also stated in the manuscript. However, the findings on Per-Chl energy transfer and quantum coherence between Per and Chl cannot be generalized for light harvesting in PCP as a whole, because the other three Per's (Per621-623) are completely ignored. If additional experiments are not possible, the following points should at least be discussed somewhere in the manuscript.

a. For example, it has been suggested that possible delocalization of Per excited states (see Bricker et al. JPC B, 2015, 119, 5755) due to strong coupling between Per's (as also seen in the SI of this work) may play a role in energy transfer in PCP, which cannot be addressed by considering only Per624.

b. Moreover, it is known that the geometry of the four Per molecules (per subunit) differs due to protein electrostatic environment (partially shown also in the SI, Figure S2-3). This difference may lead to different photophysical behavior of the individual Per molecules, which may then lead to different energy transfer dynamics to the two Chls. This also cannot be addressed by considering only Per624.

c. In this context, I also find the title of the manuscript to be slightly misleading (too general), because it is not addressed in the current manuscript whether the observed Car-Chl coherence is ubiquitous in the energy flow pathways in PCP, or selective to the Per624-Chl602 pair directly interrogated.

Minor comments:

1. A couple typos:

1) Abstract, line 13: peridin → peridinin

2) Page 9, first sentence of the third paragraph: forth → fourth

2. Page 9, second paragraph: references should be cited in the first sentence "To check for the presence of electronic coherences between Per and Chl states, whose relevance has been recently suggested in the literature". For example: Ghosh et al. J Phys Chem Lett 2017, 8, 463, and ref (23)

3. Figures S19-S21: I am assuming that these are Fourier maps from data measured (simulated) with laser spectra 1-3, but the legends are all identical. Clear labels should be added either in each figure or the figure legend.

Reviewer #3 (Remarks to the Author):

In their manuscript "Coherence in carotenoid-to-chlorophyll energy transfer", Meneghin et al. use 2D electronic spectroscopy as well as a detailed kinetic analysis and quantum mechanical calculations to explore which of different possible peridinin to chlorophyll energy path ways are actually dominating in the special light-harvesting complex PCP that contains almost only carotenoids. In particular, the authors aim at clarifying if the light-harvesting energy transfer in PCP is occurring mainly from the allowed peridinin state Per S2 either to the higher chlorophyll state Qx or the lower chlorophyll state Qy or even rather from the optically forbidden peridinin state Per S1 to the chlorophylls. The experimental approach chosen by the authors is definitively very interesting in order to answer such questions.

I have the following comments that are not necessarily ordered by importance but rather by occurrence in the manuscript:

Abstract: This is not a major point but I find statements such as „The ultrafast optical response, analyzed with state-of-the-art data analysis methods and supported by multiscale quantum chemical calculations allowed a complete mapping of the multiple pathways mechanism of energy flow in PCP and the relative time constants.” in the abstract not very concise – how about something like “Together with a global kinetic analysis and multiscale quantum chemical calculations these data allowed us to comprehensively address the contribution of all potential pathways of energy flow in PCP and the relative time constants.” or so.

In the introduction the authors state: “Dynamics of energy transport in light-harvesting complexes is traditionally attributed to the peculiar properties of tetrapyrrole compounds, mainly chlorophylls (Chls) and bacteriochlorophylls. However, exceptions are possible and they often present even more intriguing features.” : It is true that tetrapyrrole compounds dominate the light-harvesting and energy transport in most photosynthetic organisms but the way the authors describe it here leaves the impression that the contribution of carotenoids is rather an exception. In fact, they always contribute significantly to light-harvesting in the blue spectral range in almost all light-harvesting complexes but of course to a less extent than in PCP. This should be clarified.

In the introduction the authors also state: “These experiments led to models in which the excitation energy was mainly funneled from the S1 state of Per to the Qy state of Chl through the Förster mechanism, with a characteristic time constant of 2.5 - 3.5 ps.”: I did not re-read reference 8 cited here but the authors should double check if fast energy transfer through S2 hasn't been discussed

previously also for PCP and if not more complex energy transfer mechanism than Förster EET should be/have been discussed in the context of the optical forbidden state S1.

In the introduction the authors also state: "Femtosecond spectroscopic experiments have also suggested the presence of ultrafast channels of EET from S2 to the Q bands of Chl4,7-9,22 (Qx and/or vibronic states of Qy), also with possible contributions of coherent pathways." At least for other light-harvesting complexes than PCP there is convincing evidence at least for non-coherent EET from S2 to the Q bands of Chl. I know that the authors only want to address PCP but this sentence could leave the impression that there are in general doubts about this also for other light-harvesting complexes. They should clarify this.

Results: It is a clever idea to guide the specifics of the various possibilities of ultrafast experiments by prior simulation of the excitonic spectra of Pers. Also, the agreement between the calculated and experimental PCP spectrum is convincing. However, the authors claim already in the second paragraph in the results section "The emerging picture is that EET from S2 towards Qy state is much faster than towards Qx state, due to the larger couplings between S2 → Qy states. This result strongly suggests that the EET S2 → Qx channel is unlikely to show a noteworthy efficiency, disproving what was suggested in previous works" just from the agreement of the calculated and experimental PCP spectrum. This is too premature at this place in the manuscript without any other experimental results or details from the calculation and should be left more for the later sections where the data supporting this are actually discussed.

The three spectra of laser excitation used for the 2D experiment were reasonably chosen in order to dissect pure Chl signals from signals containing information from Per-Chl-interactions (Figure 1). Also the use of the assignments A-F in Figure 3 is very useful to get a quick overview of the discussed data. However, it is a little confusing that, for example, C is only shown in Figure 3 a but assigns a peak only visible in Figure 3 b/c. I guess the authors wanted to avoid too many markers in all subpanels but they should think about improvements in their way to assign the different peaks.

The authors state that the conditions that detect pure Chl signals ("laser1") in PCP "reveal a behavior in full agreement with previous 2DES characterization of Chl Solutions". Did the authors reproduce such Chl solution data using exactly their excitation condition "laser1"? If yes, it would be useful to show these data in direct comparison to the PCP data.

As one of the most important arguments against a Per S2-→QX transfer the authors state "Indeed, if S2 → Qx was the dominant pathway, the 2DES map would display a rising cross-peak at coordinates(17500,16500) cm⁻¹, but no rise at these coordinates was detected." It would be helpful to assign also this hypothetical point in Figure 3 to facilitate evaluation of this argument. The

increasing signal at position E is a convincing indication for the Per S2->Qy transfer. This signal together with the beating analysis is also a strong indication for the presence of coherent transfer, which is probably one of the most exciting findings supported by the study.

However, I would be hesitant to exclude entirely an important contribution from Per S2->Qx transfer from the absence of a signal at (17500,16500) cm⁻¹ in their data as also the signals at E are not very strong. I believe it is not necessary to stress this conclusion too much in their discussion as well as the abstract as the new insights given by this 2D-study of PCP are still interesting enough. How about stating rather something like “ These data strongly support dominant Per S2->Qy EET instead of Per S2->Qx EET, as it was generally assumed before.” in the abstract instead of “In contrast to general assumptions, the flow of energy from peridinin to chlorophyll is dominated by channels always involving the Qy state of chlorophyll as acceptor and the S2 state of the carotenoid as donor.”

I would also be hesitant putting too much emphasis on the conclusion that the Per->Chl EET is rather dominated by Per S2->Chl than Per S1->Chl. (“According to the kinetic model used for the interpretation of the 2DES data, >67% of the initially prepared S2 population is directly transferred to Qy state before Per relaxes to S1 state.”) Although the kinetic model is done accurately newer experimental data might modify the estimated S2/S1 EET balance from >67% to below 50%. Again, I recommend rather statements such as “Our data rather support dominant Per S2->Chl EET instead of Per S1->Chl EET, as it was generally assumed before”.

In summary, the study conducted by the authors is very important as 2D-spectroscopy of PCP can clarify some of the very important questions about the complex interactions between carotenoids and chlorophylls. However, in the present form the manuscript is written in a way that I find sometimes somewhat misleading in the introduction. I hope the suggestions above help to improve this. In general the paper lacks also a clear differentiation between the specific interactions and energy pathways in PCP from what can be concluded for other complexes. In the experimental section and the abstract I recommend to be more reluctant with the conclusions given the weak signals involving the S2 state – Their data strongly support Per S2->Chl Qy EET dominating over Per S2->Chl Qx EET and their analysis support Per S2->Chl EET dominating over Per S1->Chl EET but probably do not entirely exclude other pathways and certainly not for other complexes than PCP.

Reply to Reviewer Comments on Nature Communications Manuscript NCOMMS-18-06902.

Title: *Coherence in carotenoid-to-chlorophyll energy transfer*

Authors: *Elena Meneghin, Andrea Volpato, Lorenzo Cupellini, Luca Bolzonello, Sandro Jurinovich, Vincenzo Mascoli, Donatella Carbonera, Benedetta Mennucci, Elisabetta Collini**

We thank all reviewers for their detailed comments that helped us improve the clarity of the presentation in the revised version of our work. We have addressed all points in detail as listed below, and we have modified the manuscript accordingly.

Reviewers' comments in this reply letter are printed in black, and our response in blue. All modifications are also visible in a marked-up version of the manuscript and the Supp Info file. Whenever page numbers are provided in our reply, they refer to the revised, marked-up version.

We hope that with these extensive responses and modifications, our manuscript can now be recommended for publication.

On behalf of the authors
Elisabetta Collini

Reviewer #1 (Remarks to the Author):

General Impression

This is a well-written manuscript with a very careful and thorough analysis, which demonstrates the huge amount of physical insight gained from interplay of computational and careful experimental work. The figures were extremely clear and well labelled, supporting the claims made in the main text.

Main Idea(s) / Summary

A combination of careful experimental execution and global fitting analysis of 2DES spectra obtained using three different laser bandwidths with quantum chemical and quantum dynamical simulations elucidated electronic energy transfer (EET) within peridinin – chlorophyll-a – protein (PCP). It was found that EET from peridinin (Per) to chlorophyll-a (Chl-a) mainly involves the collapse of the electronic coherence between the Per S₂ and Chl-a Q_y states, in which more than two-thirds of the initially prepared Per S₂ is transferred. This electronic coherence dephases on sub-30 fs timescale and is deduced from a global analysis of experimental data. This conclusion debunks prior observations that the dominant energy transfer pathway is the Per S₁ Chl-a Q_y state.

Authors' reply: We thank the Referee for the supportive comments on our work, and we are glad that the central message of the manuscript could be fully captured.

Discussion / Critical Analysis / Comments for the Authors

1. I like how you have used time-frequency analysis to tease out the 1900 cm⁻¹ electronic coherence from signals around the below-diagonal Per S₂/Chl-a Q_y cross peak obtained with laser 3. May you elaborate further on the difficulty of applying the same analysis to the corresponding above-diagonal cross peak, since the time-frequency analysis should be able to tease out the Per vibrational mode contribution?

Authors' reply: Indeed, the time-frequency analysis has been applied also to above diagonal cross-peak region. The interpretation of the results at those coordinates is, however, a bit more challenging because the overwhelming contribution of the vibrational modes of Per almost completely hides the presence of the 1900 cm⁻¹ component.

We report below a Figure showing the time-frequency analysis performed on the position where the Q_y/S₂ above diagonal cross peak should appear. As anticipated, the oscillating behavior at this position is dominated by the strong contribution of Per vibrational coherences at 1165, 1230 and 1565 cm⁻¹. At this position, the presence of a broad and quickly damped component around 2000 cm⁻¹ can only be guessed. Nevertheless, there is an important indirect trace of the presence of such component, i.e., the appearance of negative artifacts. As commented in the main text and in Section S4.3 of the SuppInfo, negative amplitudes are generated only at frequency positions where two independent components are superimposed. These artifacts last as long as both components are active and indeed the negative features in the figure below are quickly damped. Even if indirect, this evidence supports our attribution.

We added the figure reported below also in Figure S26 and, on page 9 of the main text we added a sentence to comment this behavior.

Figure S26. Bilinear time-frequency transform of the oscillating part of the rephasing signals recorded with laser 3 and extracted at coordinates corresponding to the Q_y/S_2 above diagonal cross-peak.

2. Shouldn't the electronic coherence be closer to 2250 cm^{-1} , since the lowest Per $S_0 - S_2$ transition occurs around 17250 cm^{-1} and the Chl-a Q_y transition at 15000 cm^{-1} from the 2DES spectra shown in Figure 3 and Figure S13?

Authors' reply: The estimate of the Referee is right. Nevertheless, it is important to remember that a precise estimate of the transition energies is not fully possible just looking at the 2D maps. On the one hand, in the 2D maps the maxima of the different features are the result of a convolution of the signals with the exciting bandwidth, and therefore they could be slightly shifted with respect to their effective position. On the other hand, it is even more difficult to precisely pick the maximum of the broad signal resulting from the FT of the quickly damped oscillation component. This is an intrinsic problem connected with the time-frequency indeterminacy principle. The value of 1900 cm^{-1} was estimated through fitting of the broad signal reported in Figure 6c (purple trace). Figure 6c also shows that the broad feature mentioned above changes with the excitation frequency, as expected for an electronic coherence. With a broader bandwidth excitation, the maximum would probably be shifted at higher values, as indeed guessed by the Referee. We changed the misleading caption of Figure 6 highlighting with an 'about' that the reported 1900 cm^{-1} value is a rough estimate of the central frequency of the broad signal.

3. Is there a reference or citation for the quantum chemical simulations that supports eliminating the presence of the relaxed twisted configuration of Per in the S_2 state?

Authors' reply: The accuracy in predicting the geometry of the Per is an essential prerequisite in order to properly simulate its ground and excited states through quantum chemical approaches. Unfortunately, in highly conjugated systems such as Pers, the geometries from the crystal structures cannot be accurate enough even when the resolution is high: a resolution of 1-2 Å, in fact, cannot be enough to give accurate bond lengths and angles. Despite that, in the literature crystal structures have been largely used in combination with QM descriptions, thus leading to possible artifacts. Precisely to avoid such artifacts, in our multiscale simulations we have explicitly relaxed the geometry of each Per in the corresponding local environment using a QM(DFT)/MM strategy. For two selected Pers (614 and 624), the same strategy has also been applied to the excited state of interest. Both the ground state and the excited state optimizations have not indicated the presence of twisted configurations, but indeed distortions from the planar structures are found. The equilibrium geometry of the excited state differs from that of the ground state mainly in bond lengths and methyl rotations, whereas the backbone dihedral angles remain virtually unchanged. The same was found for Per in acetonitrile solvent, [Knecht S., et al. *J. Phys. Chem. B* 117, 44, 13808-13815; doi: 10.1021/jp4078739] where the main differences between the S_0 and S_2 equilibrium geometries were related to the bond lengths of the conjugated backbone. This strongly disproves a barrierless twisting around a double bond of the Per, even without the protein scaffold. A fast relaxation to a twisted configuration seems therefore highly unlikely.

Reviewer #2 (Remarks to the Author):

This manuscript presents a combined experimental and theoretical study on energy transfer dynamics in peridinin-chlorophyll a-protein (PCP), focusing in particular on the energy flow from the bright S₂ state of peridinin (Per), the principal light harvester in PCP, to the low-lying Q states of chlorophylls (Chls). By using ultrafast two-dimensional electronic spectroscopy, the authors identify a Per S₂-Chl Q_y cross peak with a 63 fs energy transfer rate, proposing the S₂→Q_y pathway as the dominant energy transfer pathway in PCP instead of S₂→Q_x, as has been reported in the literature. In addition, signatures of potential electronic coherence between Per S₂ and Chl Q_y are observed, which appears consistent with the observed very rapid (63-fs) energy transfer rate.

PCP, found in the light-harvesting machinery of marine dinoflagellates, exhibits distinct photophysical properties from the light-harvesting complexes of bacteria and plants: 1) it uses carotenoids (peridins), not chlorophylls, as the principal light harvester, and 2) peridinin is a highly substituted carotenoid, giving rise to unusual behaviors, e.g. strong dependence on solvent polarity, presence of an intramolecular charge transfer state. These unique characteristics of PCP make it an intriguing system to elucidate the dynamics of Car-Chl energy transfer, specifically the photophysical role of peridins, which is still not completely known. Furthermore, the experiments employ advanced spectroscopic technique and are paired with cutting-edge theoretical work. However, the two major conclusions (direct S₂->Q_y energy transfer and role of coherence in energy transfer) both require further substantiation, as described in more detail below. Without additional conclusions or justification, this manuscript is more appropriate for a field-specific journal.

Authors' reply: Before answering point by point to the Referee's comments, we would like to report a general comment.

We understand that the Referee has two main concerns:

- 1) About the claim that energy transfer from Per to Chl follows mainly the S₂-Q_y channel, with little or no involvement of Q_x states, as it was previously suggested.
- 2) About the generality of our claim (and our title)

We realize that these concerns probably arose from a description and a discussion of the data that were not clear enough in the original version of the manuscript. To avoid any possible further misunderstanding, we have added some new parts and revised others as detailed below.

In particular, as it regards claim 1, we added in the SuppInfo additional information substantiating our interpretation (see the answer to points 1a and 1b below). Also following the suggestions of Referee #3, we tried to generalize our conclusion, leaving open the possible presence of alternative mechanisms, although the data suggest they would represent only minority contributions.

As it regards claim 2, we agree with the Referee that we are addressing a specific protein and that our data can capture only the transfer from the red-most states of Pers to Chl. However, our main conclusion is to demonstrate that the transfer Car to Chl can sustain coherence dynamics. In other words, we are not implying that all the Car-to-Chl transfers are following coherent dynamics, but that, in principle, this is possible, also when, in the first instance, the spectral detuning seems to be not entirely favorable.

This is an important and general finding in the attempt of unveiling the still unclear mechanism of energy flow from carotenoid to chlorophyll. The unambiguous detection of this mechanism has been possible in PCP thanks to its relatively simple pigments content and ratio, which allowed isolating the coherent S₂-Q_y channel from the rest of the dynamics. In other structures characterized by Car-Chl transfers, such as CP29 and LHCII, the presence of different kinds of carotenoids and chlorophylls and the higher number of pigments involved in the overall dynamics make the clear identification of this channel more challenging.

Major comments:

1. Page 5, last paragraph: the authors state that they observe no S₂-Q_x cross peaks, which leads them to propose Per S₂→Chl Q_y as the exclusive energy transfer mechanism between Per and Chl. This is reasonable given the significantly smaller coupling between S₂-Q_x compared to S₂-Q_y, but also surprising considering the poor spectral overlap as well as larger energy gap between Per S₂ and Chl Q_y. Furthermore, if energy transfers from S₂ to Q_x on similar or slower timescale than from Q_x to Q_y, no peak would appear. Thus, a more detailed analysis might be worthwhile, such as:

a. To rule out inverted kinetics as described, the authors should show the decay of S₂ matches the rise of Q_y.

Authors' reply: A first demonstration that indeed the rise of Q_y corresponds to the decay of S_2 is provided in Figure S17 (now S18), which shows the amplitude map of the ultrafast component of 63 fs obtained by the kinetic global fitting. The amplitude map confirms that the diagonal signal at about $(17500, 17500) \text{ cm}^{-1}$ (corresponding to dynamics of the S_2 population) and the cross peak below diagonal (corresponding to the S_2 - Q_y transfer at $(17500, 14850) \text{ cm}^{-1}$) grow with the same dynamics.

To make the maps in Figure S18 more easily readable, we also extracted the 1D traces at the aforementioned coordinates as a function of t_2 , and we prepared the figure below, also added in SuppInfo as Figure S14. It is clear that the early time behavior of the two traces is nicely matching.

Now, the fact that the signal at $(17500, 17500) \text{ cm}^{-1}$ appears as a growing and not as a decaying trace, is because the signal at this position is dominated by ESA from S_2 to S_n rather than by SE or GSB. ESA produces negative signals and, therefore, as the S_2 population decays, the corresponding ESA feature becomes less negative and overall the trace is growing.

This is the same reason why also in Figure S18(a) the peaks centered at excitation energy 17500 and 14850 cm^{-1} have the same negative sign.

In order to have a more direct insight into the decay of S_2 population and growing of Q_y population, the simulated data can be of great help, since, as stated on page 10, the simulations do not account for ESA towards higher energy states, and therefore they allow isolating more clearly the contribution of S_2 population decay. The simulated data are reported in panel (b) of the Figure S14.

Again, the early time dynamics corresponding to the decay of S_2 population and the growing of Q_y population are matching.

Figure S14. Evolution of the pure absorptive signal recorded with laser 3 along population time at selected excitation/emission energies. Experimental (a) and simulated (b) data extracted at coordinates corresponding to the dynamics of the S_2 population (blue) and at the cross peak below diagonal corresponding to the S_2/Q_y transfer (red). Thick lines in panel (a) represent the fitting traces. The simulated data (blue trace in panel (b)) do not account for the $S_2 \rightarrow S_n$ ESA contribution.

b. To rule out the presence of an S_2 peak, while it clearly looks like the major feature is the intense negative $S_1 \rightarrow S_n$ excited-state absorption (ESA) of Per624 that is masking all the positive features, if there is a visible S_2 - Q_x energy transfer channel, monitoring the difference in the waiting time traces between where the S_2 - Q_x peak should show up and where there is only Per S_1 ESA may unmask the kinetics of S_2 - Q_x energy transfer.

Authors' reply: Following the suggestion of the Referee, we tentatively compared 1D traces extracted at coordinates where the S_2 - Q_x peak should show up and where only S_1 - S_n contribution is expected. This basically means to compare traces along a vertical line at the same excitation energy (17500 cm^{-1}).

The dynamics at these coordinates is quite complex because several processes giving rise to signals with opposite signs are superimposed ($S_2 \rightarrow S_n$, $S_1 \rightarrow S_n$, $S_2 \rightarrow S_1 \rightarrow S_0$, etc.) Moreover, it is not straightforward to pick precisely the coordinates where S_1 and Q_x contributions are expected because of the null or negligible dipole moment of the two states. Besides, the broadening of the peaks at room temperature is not helping in this task. Keeping this in mind, in the Figure below we compare traces extracted for excitation energy 17500 cm^{-1} at increasing values of emission energy. The S_2 - Q_x peak should appear close to coordinates $(17500, 16150) \text{ cm}^{-1}$, red trace.

Moving from higher to lower values of the emission coordinate (i.e. going from blue to green traces in the Figure), at early times we observe a clear contribution of $S_2 \rightarrow S_n$ (rise of a negative signal in blue traces, as explained in the previous point) which is progressively cancelled out at lower energies by the $S_1 \rightarrow S_n$ process (green traces).

The comparison of the 1D traces shown in the figure below does not highlight any explicit additional dynamics that could be attributed to an $S_2 \rightarrow Q_x$ transfer. While the presence of such a channel cannot be completely ruled out because of the aforementioned complex dynamics, our data suggest that, even if present, this channel is undoubtedly negligible with respect to others because we cannot isolate from the overall dynamics an additional ultrafast component attributable to $S_2 \rightarrow Q_x$.

These experimental findings are also supported by theory, predicting much lower rates for the $S_2 \rightarrow Q_x$ transfer. Basing on that, we performed an additional calculation solving a modified version of the kinetic model presented in the main text. Two channels involving Q_x state were added: $Q_x \rightarrow Q_y$, with a rate of $1/(150 \text{ fs})$, fixed in agreement with [Polivka et al., Archives of Biochemistry and Biophysics 458 (2007), pp. 111-120; doi: 10.1016/j.abb.2006.10.006], and $S_2 \rightarrow Q_x$. The time constant associated with this last process is a crucial parameter to calculate the maximum population reached on Q_x because it defines how competitive is the channel in the ultrafast branching of the S_2 population decay. This parameter was chosen in agreement with the quantum mechanical calculations, which provide a distribution of rates for $S_2 \rightarrow Q_x$ with an average time constant of 1000 fs in the red tale region of the peridinin band, see Figure S4. Within this expanded model, the maximum fraction of Q_x population cascading from the S_2 state is estimated lower than 2.5%.

Additional evidence for a direct $S_2 \rightarrow Q_y$ pathway is the presence of a coherent dynamics between the two states, as witnessed by the quickly damped oscillation at about 1900 cm^{-1} . If the transfer from S_2 to Q_y was mediated by an intermediate state (Q_x or S_1), then we would not have recorded such coherent dynamics.

In summary, we believe that this wealth of experimental pieces of evidence, together with the results of calculations, rule out a relevant involvement of Q_x in the transfer mechanism, at least in the considered time and spectral window.

Pure absorptive signal as a function of population time. Excitation energy = 17500cm^{-1} (i.e. S_2 energy). The traces are extracted at increasing values of emission energy. The trace extracted at coordinates where the putative S_2 - Q_x dynamics should show up is highlighted in red.

2. The authors describe the ~ 22 fs processes observed as arising from coherence between Per S_2 and Chl Q_y . In the case of ultrafast excitation, however, the defined phase relationship between pulses one and two excites a coherence, which decoheres rapidly (< 50 fs timescale) due to the small couplings between the pigments. These oscillations emerge from ultrafast excitation and carry no information about energy transfer dynamics. These behaviors are described theoretically in the literature, for example, Schlau-Cohen, et al. *Nature Chem* 2012 4, 389.

Authors' reply: We absolutely agree with the Referee, and we would like to stress that it is crucial for us to distinguish between the 'coherence of the excitonic state' that we prepare with ultrashort excitation and the 'coherence of the process', where coherence dynamics is functional to the transfer. Indeed, when an excitonic system is excited by a broadband ultrashort laser pulse, like in our case, a coherent superposition of states can be generated, which then begins to oscillate. As reported in the cited *Nature Chem* paper, these oscillations are just the manifestation of the coherent nature of the excitonic states prepared by ultrafast excitation. The stronger the coupling, the longer the dephasing time. We observed similar behavior for example in excitonic J-aggregates where the strong coupling guaranteed a dephasing time of about 200 fs (Bolzonello L. et al., *J. Phys. Chem. Lett.* 2016, 7, 4996–5001; doi: 10.1021/acs.jpcclett.6b02433).

The situation here is different because we could demonstrate that the dephasing of a coherent superposition of states is functional to the transfer. And this interpretation is based not only on the analysis of the oscillatory behavior (the component at about 2000cm^{-1} found at S_2 - Q_y cross peak position), but also on the non-oscillating population dynamics. We indeed captured a rise at diagonal position ($14850, 14850\text{cm}^{-1}$) (Figure 4b), corresponding to Q_y population signal. This rise can be justified only with a coherent mechanism of energy transfer (as illustrated in Figure 4a) where the dephasing of the S_2 - Q_y coherence drives population on Q_y .

3. A final major issue I have with this work is that only one specific energy transfer pathway (Per624→Chl602) is directly interrogated due to limited bandwidth of the NOPA spectrum. This is only a single pair of Per and Chl out of the many donor acceptor pairs that can be formed with 4 Per's and one Chl per subunit of PCP. The NOPA spectra employed in this work may be beneficial in that they allow for selective (de)excitation of Per624, as also stated in the manuscript. However, the findings on Per-Chl energy transfer and quantum coherence between Per and Chl cannot be generalized for light harvesting in PCP as a whole, because the other three Per's (Per621-623) are completely ignored. If additional experiments are not possible, the following points should at least be discussed somewhere in the manuscript.

a. For example, it has been suggested that possible delocalization of Per excited states (see Bricker et al. JPC B, 2015, 119, 5755) due to strong coupling between Per's (as also seen in the SI of this work) may play a role in energy transfer in PCP, which cannot be addressed by considering only Per624.

b. Moreover, it is known that the geometry of the four Per molecules (per subunit) differs due to protein electrostatic environment (partially shown also in the SI, Figure S2-3). This difference may lead to different photophysical behavior of the individual Per molecules, which may then lead to different energy transfer dynamics to the two Chls. This also cannot be addressed by considering only Per624.

Authors' reply: We thank the Referee for this comment that gave us the chance of improving the clarity of the discussion reported on page 3 (in the Section 'Prediction of excitonic spectrum and rate analysis').

First of all, it is important to highlight that what the experiment is able of capturing is the transfer between the red-most states of Pers and Q bands of Chls.

A simple physical intuition based on the estimate of the energy detuning between donor and acceptor states, suggests that the red-most Per states are also the ones that should guarantee the fastest rates of transport. Surely, there can be other possible channels, involving higher energy states of Pers which cannot be captured in our experimental conditions, but we expect these other channels to be anyway slower and less relevant in the overall transfer kinetics (as also confirmed by calculations).

That said, the experimental data we are presenting are absolutely general, in the sense that we are characterizing the energy transport from red Pers to Chl in PCP. In the main text, indeed, we comment on the mechanisms of energy transfer deduced from the experiment always referring generally to Per and Chl states (not to Per624 and Chl602 states).

It is certainly true that the experiment is not able of labeling the involved states, and it cannot attribute the investigated states to a specific molecule of Per or Chl in PCP. This can, however, be done with the help of QM calculations.

As we reported on page 3 and in Section S1-3 of the SuppInfo, the electronic properties of the full ensemble of chromophores have been calculated starting from an exciton Hamiltonian that includes the (S_2) bright states of the eight Pers, and the Q (Q_y, Q_x) and Soret (B_y, B_x) pair of states of the two chlorophylls (Chls), resulting in 16 total excitonic states. Therefore, to answer to the point (a) of the Referee, our analysis is not limited to a single Per, instead it considers delocalized states, which are solutions of the full Hamiltonian. We also stress that all these excitonic properties are calculated by including both electrostatic and polarization effect of the protein using an atomistic MMPol model.

Starting from this model, which allows delocalization over all Per molecules, our calculations clearly show (see for example Figure S4) that the red-most states of the Pers ensemble (i.e., the ones involved in the experiment), although maintaining an excitonic and partially delocalized character, become virtually localized on a single Per. Of course, this localization is not always on Per624 but, depending on the disorder, it can involve other Pers. This has been written on page S9:

"The Per -> Chl EET essentially occurs from a single Per donor, which may be Per624, 611(621) or 613(623) depending on which is the red-most Per in a particular configuration of static disorder."

This is precisely what the referee suggests in point b. With localization, the spectral overlap to the Q_y states increases because delocalization suppresses the diagonal exciton-phonon coupling: for this reason, delocalized states are actually less efficient in transferring energy to the Q_y states.

Only when we simulated the 2D response to compare with experiments directly, we explicitly referred to Per624 and Chl602. We chose to consider only this dimer for the sake of simplicity, because calculations suggested that (i) the red-most states are localized on one specific Per, (ii) all considered, the fastest transfer occurs from Per624 to Chl602, and (iii) the transition dipole moment of Per624 is parallel to that of Chl602, which enhances the cross-peak in an all-parallel configuration.

We now realize that the whole simulation strategy was not clearly explained in the main text. To avoid further misunderstandings, we modified the main text on page 10 as follows:

"[...] here, however, only the excited states Q_y of Chl602 and S_2 of Per624 were considered for the sake of simplicity. Calculations indeed suggested that this pair provides the major contribution to the faster EET channel captured by the experimental data and therefore this choice allows simplification of the calculations without losing in generality."

c. In this context, I also find the title of the manuscript to be slightly misleading (too general), because it is not addressed in the current manuscript whether the observed Car-Chl coherence is ubiquitous in the energy flow pathways in PCP, or selective to the Per624-Chl602 pair directly interrogated.

Authors' reply: In the light of what stated before, we hope we convinced the Referee that what we are presenting is general enough to deserve the originally proposed title.

Minor comments:

1. A couple typos:

1) Abstract, line 13: peridin → peridinin

2) Page 9, first sentence of the third paragraph: forth → fourth

2. Page 9, second paragraph: references should be cited in the first sentence "To check for the presence of electronic coherences between Per and Chl states, whose relevance has been recently suggested in the literature". For example: Ghosh et al. J Phys Chem Lett 2017, 8, 463, and ref (23)

3. Figures S19-S21: I am assuming that these are Fourier maps from data measured (simulated) with laser spectra 1-3, but the legends are all identical. Clear labels should be added either in each figure or the figure legend.

Authors' reply: We are grateful to the Referee for pointing out these problems, promptly corrected in the revised versions of the files.

Reviewer #3 (Remarks to the Author):

In their manuscript “Coherence in carotenoid-to-chlorophyll energy transfer”, Meneghin et al. use 2D electronic spectroscopy as well as a detailed kinetic analysis and quantum mechanical calculations to explore which of different possible peridinin to chlorophyll energy path ways are actually dominating in the special light-harvesting complex PCP that contains almost only carotenoids. In particular, the authors aim at clarifying if the light-harvesting energy transfer in PCP is occurring mainly from the allowed peridinin state Per S2 either to the higher chlorophyll state Qx or the lower chlorophyll state Qy or even rather from the optically forbidden peridinin state Per S1 to the chlorophylls. The experimental approach chosen by the authors is definitively very interesting in order to answer such questions.

I have the following comments that are not necessarily ordered by importance but rather by occurrence in the manuscript:

Abstract: This is not a major point but I find statements such as „The ultrafast optical response, analyzed with state-of-the-art data analysis methods and supported by multiscale quantum chemical calculations allowed a complete mapping of the multiple pathways mechanism of energy flow in PCP and the relative time constants.” in the abstract not very concise – how about something like “Together with a global kinetic analysis and multiscale quantum chemical calculations these data allowed us to comprehensively address the contribution of all potential pathways of energy flow in PCP and the relative time constants.” or so.

Authors’ reply: Thanks for the suggestion, readily applied!

In the introduction the authors state: “Dynamics of energy transport in light harvesting complexes is traditionally attributed to the peculiar properties of tetrapyrrole compounds, mainly chlorophylls (Chls) and bacteriochlorophylls. However, exceptions are possible and they often present even more intriguing features.” : It is true that tetrapyrrole compounds dominate the light-harvesting and energy transport in most photosynthetic organisms but the way the authors describe it here leaves the impression that the contribution of carotenoids is rather an exception. In fact, they always contribute significantly to light-harvesting in the blue spectral range in almost all light-harvesting complexes but of course to a less extent than in PCP. This should be clarified.

Authors’ reply: The sentence says that the exception is PCP (because of the unusual Chl: carotenoids ratio). Indeed, we did not want to imply that the use of carotenoids in light harvesting is an exception. To avoid confusion, the cited sentence on page 2 has been modified as follows:

“Dynamics of energy transport in light-harvesting complexes is traditionally attributed to the peculiar properties of tetrapyrrole compounds, mainly chlorophylls (Chls) and bacteriochlorophylls. A particular family of marine organisms of the dinoflagellates phylum has instead evolved a light-harvesting machinery, mainly relying on carotenoids (Cars) as major harvesting pigments.”

And a bit later we added:

“PCP, with its high content of carotenoids, represents therefore an ideal test case. Moreover, the presence of peridinins, peculiar carotenoids substituted with polar groups, may lead to intriguing features.”

In the introduction the authors also state: “These experiments led to models in which the excitation energy was mainly funneled from the S1 state of Per to the Qy state of Chl through the Förster mechanism, with a characteristic time constant of 2.5 - 3.5 ps.”: I did not re-read reference 8 cited here but the authors should double check if fast energy transfer through S2 hasn’t been discussed previously also for PCP and if not more complex energy transfer mechanism than Förster EET should be/have been discussed in the context of the optical forbidden state S1.

Authors' reply: Reference 8 was cited to confirm the timescale of the S_1 - Q_y transfer (In Ref. 8 the authors report: "The measured S1/ICT lifetime of peridinin in MFPCP [main form PCP] varies from 2.5 to 3.5 ps, significantly shorter than for peridinin in any solvent").

We realize that the sentence was a bit too hasty because we did not want to imply any claim on the nature of the transfer from the S_1 state, still object of debate in the literature, but just highlighting the different timescales involved. We thank the referee for signaling us this inaccuracy.

To avoid any misunderstanding, we modified the sentence by removing the explicit reference to a Forster mechanism:

"These experiments led to models in which the excitation energy was mainly funneled from the S_1 state of Per to the Q_y state of Chl with a characteristic time constant of 2.5 - 3.5 ps⁸."

For the sake of discussion, the review in Ref. 8 suggests two possible mechanisms of energy transfer from S_1 to Q_y : the first is a Forster mechanism and the second is a "mechanism that depends on the density of acceptor states". Nevertheless, the authors of the review lean towards the first hypothesis since they cite the work of Polivka et al. [Photosynthesis Research (2005) 86: 217–227; doi: 10.1007/s11120-005-1447-x] to demonstrate that the energy transfer rate is a function of the spectral overlap between the peridinin (donor) emission spectrum and the acceptor absorption spectrum. In the aforementioned work, the dependency of energy transfer rate on the spectral overlap of donor and acceptor is demonstrated by changing the nature of the acceptor in reconstituted PCP: Chla, acetylChla, Chlb, Chld, and BChla.

Other works support the Forster mechanisms:

- Krueger et al. Biophysical Journal 2001, 80(6), 2843–2855; doi: 10.1016/S0006-3495(01)76251-0;
- Zigmantas et al. PNAS 2002, 99, 26, 16760-16765; doi: 10.1073/pnas.262537599;
- Ghosh et al. J. Phys. Chem. Lett. 2017, 8, 463–469; doi: 10.1021/acs.jpcllett.6b02881;

The review only quickly mentioned the possibility of a different energy mechanism insensitive to spectral overlap, as suggested by Papagiannakis et al. [Biochemistry, 2004, Vol. 43, No. 49, 15303-15309; doi: 10.1021/bi047977r]. The work concludes that S_1 and ICT states are separated excited states in peridinin and ICT state act as the main donor to Chla.

In summary, the nature of S_1 /ICT state in Per and also the mechanism of the transfer from this state to Chl is still intensively debated. The discussion about the nature of this transfer channel, however, goes beyond the scope of this manuscript.

In the introduction the authors also state: "Femtosecond spectroscopic experiments have also suggested the presence of ultrafast channels of EET from S_2 to the Q bands of Chl^{4,7–9,22} (Q_x and/or vibronic states of Q_y), also with possible contributions of coherent pathways." At least for other light-harvesting complexes than PCP there is convincing evidence at least for non-coherent EET from S_2 to the Q bands of Chl. I know that the authors only want to address PCP but this sentence could leave the impression that there are in general doubts about this also for other light-harvesting complexes. They should clarify this.

Authors' reply: We thank the Referee for highlighting the possible ambiguity between the peculiar properties of Pers and PCP and the general behavior of other antenna complexes bearing carotenoids and chlorophylls. We modified the introduction, especially the sentences cited by the Referee, to clarify that we are discussing, in particular, the behavior of PCP and that our discussion does not want to be universal. We also added a reference to a review describing general properties of Cars (now ref. 22):

"Traditionally, Cars electronic properties are described with a three-level model including a ground state, S_0 ($1^1A_g^-$), and two low-lying singlet excited states, S_1 ($2^1A_g^-$) and S_2 ($1^1B_u^+$). It follows that $S_0 \rightarrow S_1$ transition is symmetry-forbidden and that S_2 is the first bright state²². Early femtosecond spectroscopy experiments on PCP^{3–5} revealed that, after photoexcitation, the S_2 state rapidly relaxes to S_1 state via internal conversion in tens-hundreds of femtoseconds.

[...]

Like for other antenna complexes bearing carotenoids, femtosecond spectroscopic experiments on PCP have also suggested the presence of ultrafast channels of EET from S_2 to the Q bands of Chl^{4,7–9,23} (Q_x and/or vibronic states of Q_y), also with possible contributions of coherent pathways^{24,25}."

Results: It is a clever idea to guide the specifics of the various possibilities of ultrafast experiments by prior simulation of the excitonic spectra of Pers. Also, the agreement between the calculated and experimental PCP spectrum is convincing. However, the authors claim already in the second paragraph in the results section “The emerging picture is that EET from S₂ towards Q_y state is much faster than towards Q_x state, due to the larger couplings between S₂ → Q_y states. This result strongly suggests that the EET S₂ → Q_x channel is unlikely to show a noteworthy efficiency, disproving what was suggested in previous works” just from the agreement of the calculated and experimental PCP spectrum. This is too premature at this place in the manuscript without any other experimental results or details from the calculation and should be left more for the later sections where the data supporting this are actually discussed.

Authors’ reply: Probably the phrasing of the cited sentence was not clear and precise enough. Indeed, the sentence was meant to explain the results of the calculations, predicting that the rate of the transfer towards Q_y is greater than the rate towards Q_x. We modified the sentence on page 3 to make its content clearer:

“The calculations predict that EET from S₂ towards Q_y state is much faster than towards Q_x state, due to the larger couplings between S₂ → Q_y states. According to that, the EET S₂ → Q_x channel is unlikely to show a noteworthy efficiency, [...]”

The three spectra of laser excitation used for the 2D experiment were reasonably chosen in order to dissect pure Chl signals from signals containing information from Per-Chl-interactions (Figure 1). Also the use of the assignments A-F in Figure 3 is very useful to get a quick overview of the discussed data. However, it is a little confusing that, for example, C is only shown in Figure 3 a but assigns a peak only visible in Figure 3 b/c. I guess the authors wanted to avoid too many markers in all subpanels but they should think about improvements in their way to assign the different peaks.

Authors’ reply: the choice of reporting markers only on the panel (a) was to avoid the figure overcrowding and to allow a clear inspection of the important features commented on panels (b) and (c). However, we thank the Referee for letting us notice that this choice could lead to a somewhat tricky interpretation. We revised Figure 3 reporting the markers in all the panels.

The authors state that the conditions that detect pure Chl signals (“laser1”) in PCP “reveal a behavior in full agreement with previous 2DES characterization of Chl Solutions”. Did the authors reproduce such Chl solution data using exactly their excitation condition “laser1”? If yes, it would be useful to show these data in direct comparison to the PCP data.

Authors’ reply: We performed measurements on methanol solutions of Chl a in the same conditions labeled as ‘laser 1’. For the Referee convenience, we report below a figure showing the obtained results.

In the original manuscript, we did not report this Figure because the properties and the dynamics of Chl are already well characterized, also in terms of 2D spectroscopy (see for example Meneghin et al. Sci. Rep. (2017) 7, 11389; doi: 10.1038/s41598-017-11621-2, and references therein).

In the meantime, the data shown in the Figure below have also been published in Meneghin et al., Chem Phys (2018) (in press), (doi: 10.1016/j.chemphys.2018.03.003), and therefore we can now refer to this paper for further comparison (added as Ref 34 on page 4).

In the comparison of the response of Chl in methanol and in protein, it has to be recalled that the molecule has different photophysical properties in the two conditions. For example, the absorption maximum for Q_y band is at 15050 cm⁻¹ in methanol and 14850 cm⁻¹ in PCP. This is also reflected on the position of the main diagonal peak in the 2D maps.

Absorptive 2DES signal of Chl a in methanol at 295 K at selected value of population time, obtained with laser 3. Maps are normalized to 1 at their maximum.

As one of the most important arguments against a Per $S_2 \rightarrow Q_x$ transfer the authors state “Indeed, if $S_2 \rightarrow Q_x$ was the dominant pathway, the 2DES map would display a rising cross-peak at coordinates (17500, 16500) cm^{-1} , but no rise at these coordinates was detected.” It would be helpful to assign also this hypothetical point in Figure 3 to facilitate evaluation of this argument.

Authors’ reply: As suggested, we added in the panels of Figure 3 a dashed line circle indicating this position. On page 6 we also specified:

“Indeed, if $S_2 \rightarrow Q_x$ was the dominant pathway, the 2DES map would display a rising cross-peak at coordinates (17500, 16150) cm^{-1} , tentatively indicated with a dotted circle in Fig. 3(a-c), but no rise at these coordinates was detected.”

The increasing signal at position E is a convincing indication for the Per $S_2 \rightarrow Q_y$ transfer. This signal together with the beating analysis is also a strong indication for the presence of coherent transfer, which is probably one of the most exciting findings supported by the study.

However, I would be hesitant to exclude entirely an important contribution from Per $S_2 \rightarrow Q_x$ transfer from the absence of a signal at (17500, 16500) cm^{-1} in their data as also the signals at E are not very strong. I believe it is not necessary to stress this conclusion too much in their discussion as well as the abstract as the new insights given by this 2D-study of PCP are still interesting enough. How about stating rather something like “These data strongly support dominant Per $S_2 \rightarrow Q_y$ EET instead of Per $S_2 \rightarrow Q_x$ EET, as it was generally assumed before.” in the abstract instead of “In

contrast to general assumptions, the flow of energy from peridinin to chlorophyll is dominated by channels always involving the Q_y state of chlorophyll as acceptor and the S_2 state of the carotenoid as donor.”

Authors’ reply: We agree with the Referee, and indeed we followed his suggestion. Nevertheless, we would like to point out that several experimental findings suggest a negligibly small involvement of Q_x (as discussed in answer to Referee #2):

- The decay of S_2 matches the rise of Q_y , as highlighted in the new Figure S14 added to the SuppInfo.
- The comparison of 1D traces extracted at coordinates where the $S_2 \rightarrow Q_x$ peak should show up and where only $S_1 \rightarrow S_n$ contribution is expected did not highlight any evident dynamics that could be attributed to an $S_2 \rightarrow Q_x$ transfer.

- We could capture coherent dynamics between the S_2 and Q_y states, as witnessed by the quickly damped oscillation at about 1900 cm^{-1} . If the transfer from S_2 to Q_y was mediated by an intermediate state (Q_x or S_1), then we would not have recorded such coherent dynamics.

In summary, we believe that this wealth of experimental pieces of evidence, together with the results of calculations, predicting much lower rates for the S_2 - Q_x transfer, rule out a relevant involvement of Q_x in the transfer mechanism, at least in the considered time and spectral window.

I would also be hesitant putting too much emphasis on the conclusion that the Per->Chl EET is rather dominated by Per S_2 ->Chl than Per S_1 ->Chl. (“According to the kinetic model used for the interpretation of the 2DES data, >67% of the initially prepared S_2 population is directly transferred to Q_y state before Per relaxes to S_1 state.”) Although the kinetic model is done accurately newer experimental data might modify the estimated S_2/S_1 EET balance from >67% to below 50%. Again, I recommend rather statements such as “Our data rather support dominant Per S_2 ->Chl EET instead of Per S_1 ->Chl EET, as it was generally assumed before”.

Authors’ reply: we thank the Referee for helping us in generalizing our claims. We modified the text on page 10 as suggested, removing that sentence.

In summary, the study conducted by the authors is very important as 2D-spectroscopy of PCP can clarify some of the very important questions about the complex interactions between carotenoids and chlorophylls. However, in the present form the manuscript is written in a way that I find sometimes somewhat misleading in the introduction. I hope the suggestions above help to improve this. In general the paper lacks also a clear differentiation between the specific interactions and energy path ways in PCP from what can be concluded for other complexes. In the experimental section and the abstract I recommend to be more reluctant with the conclusions given the weak signals involving the S_2 state – Their data strongly support Per S_2 ->Chl Q_y EET dominating over Per S_2 ->Chl Q_x EET and their analysis support Per S_2 ->Chl EET dominating over Per S_1 ->Chl EET but probably do not entirely exclude other pathways and certainly not for other complexes than PCP.

Authors’ reply: We thank the Referee for all his/her comments that indeed helped us a lot in improving the readability and the clarity of the manuscript. We are confident that the changes we made through the text are now making more explicit where we are specifically referring to PCP and not the other complexes. Moreover, we revised the conclusions, leaving as open the possible presence of alternative mechanisms, although the data suggest they would represent only minority contributions.

Reviewer #2 (Remarks to the Author):

The additional explanation and analysis has satisfactorily addressed my concerns and suggestions. I now recommend acceptance of this manuscript.

Reviewer #3 (Remarks to the Author):

In general, the authors have addressed very carefully each comment of the reviewers. The abstract is now significantly more balanced regarding, for example, alternative interpretations or path ways other than coherent Per S₂->Chl Q_y ET. However, I am still not entirely convinced that the authors have sufficiently addressed this throughout the remaining manuscript. For example, in the conclusions still sentences exist like "The emerging picture is that, contrary to that generally assumed, the energy flow from Per to Chl is dominated by a channel involving always the Q_y state of Chl as acceptor and the S₂ state of the Per as donor. These findings finally clarify a long-term discussion about the effective states involved in the energy flow from Per to Chl.". The authors should check once more the entire manuscript for statements that might be too final/premature based on what they can actually see in their experimental/ computational data. So in summary, the authors have already put significant effort to improve the manuscript but I would ask them once more to carefully consider all remaining criticism of all referees.

Padova, 4th July, 2018

Dear Editor,

In addition to the requested editorial changes, we have modified the manuscript to address the latest request of Referee#3, who asked for more balanced statements in our discussion and conclusions. Below the point-by-point response:

Reviewer #2

The additional explanation and analysis has satisfactorily addressed my concerns and suggestions. I now recommend acceptance of this manuscript.

We thank the Referee for the additional time devoted to our manuscript and for the favorable report.

Reviewer #3

In general, the authors have addressed very carefully each comment of the reviewers. The abstract is now significantly more balanced regarding, for example, alternative interpretations or path ways other than coherent Per S2->Chl Qy ET. However, I am still not entirely convinced that the authors have sufficiently addressed this throughout the remaining manuscript. For example, in the conclusions still sentences exist like “The emerging picture is that, contrary to that generally assumed, the energy flow from Per to Chl is dominated by a channel involving always the Qy state of Chl as acceptor and the S2 state of the Per as donor. These findings finally clarify a long-term discussion about the effective states involved in the energy flow from Per to Chl.”. The authors should check once more the entire manuscript for statements that might be too final/premature based on what they can actually see in their experimental / computational data. So in summary, the authors have already put significant effort to improve the manuscript but I would ask them once more to carefully consider all remaining criticism of all referees.

We thank the Referee for recognizing our attempts to balance our statements. We went through the manuscript and we slightly modified sentences in which our statements may result too ‘premature’:

On page 2:

‘The associated timescales are characterized, and the identification of important mechanistic details, including the presence of coherent transfer and the **strong potential** involvement of vibrational modes of Per, is assessed.’

Prof. Elisabetta Collini
DIPARTIMENTO DI SCIENZE CHIMICHE
via Marzolo 1 - 35131 Padova (Italy)
tel +39 049 8275131 - Fax +39 049 8275135
elisabetta.collini@unipd.it

UNIVERSITÀ
DEGLI STUDI
DI PADOVA

On Page 8:

‘It should be noted that, in the time **and spectral** window investigated, the population of the final acceptor Q_y state, although fed by two distinct channels, derives mainly from S_2 .’

On Page 11, the sentence cited by the Referee has been changed as:

‘The picture emerging from computational and experimental data is that, contrary to that generally assumed, the energy flow from Per to Chl **in the investigated time and spectral windows**, is dominated by a direct channel involving Q_y state of Chl as acceptor and the S_2 state of the Per as donor. These findings **represent a further insight towards the full resolution of a long-term discussion** about the effective states involved in the energy flow from Per to Chl.’

And, again on page 11:

‘This is an important **and general** piece of information in the current efforts of unveiling the still unclear mechanism of energy flow from carotenoids to chlorophyll.’

Thanks and best regards
Elisabetta Collini